# Regional Development of Glioblastoma: The Anatomical Conundrum of Cancer Biology and Its Surgical Implication

**DOI:** 10.3390/cells11081349

**Published:** 2022-04-15

**Authors:** Ciro De Luca, Assunta Virtuoso, Michele Papa, Francesco Certo, Giuseppe Maria Vincenzo Barbagallo, Roberto Altieri

**Affiliations:** 1Laboratory of Neuronal Network Morphology and Systems Biology, Department of Mental and Physical Health and Preventive Medicine, University of Campania “Luigi Vanvitelli”, 80138 Naples, Italy; ciro.deluca@unicampania.it (C.D.L.); assunta.virtuoso@unicampania.it (A.V.); 2SYSBIO Centre of Systems Biology ISBE-IT, 20126 Milano, Italy; 3Department of Neurological Surgery, Policlinico “G. Rodolico-S. Marco” University Hospital, 95121 Catania, Italy; cicciocerto@yahoo.it (F.C.); giuseppebarbagal@hotmail.com (G.M.V.B.); 4Interdisciplinary Research Center on Brain Tumors Diagnosis and Treatment, University of Catania, 95123 Catania, Italy

**Keywords:** glioblastoma, neuroanatomy, biological signature, neurosurgery, diffusive, bulky, regional phenotype

## Abstract

Glioblastoma (GBM) are among the most common malignant central nervous system (CNS) cancers, they are relatively rare. This evidence suggests that the CNS microenvironment is naturally equipped to control proliferative cells, although, rarely, failure of this system can lead to cancer development. Moreover, the adult CNS is innately non-permissive to glioma cell invasion. Thus, glioma etiology remains largely unknown. In this review, we analyze the anatomical and biological basis of gliomagenesis considering neural stem cells, the spatiotemporal diversity of astrocytes, microglia, neurons and glutamate transporters, extracellular matrix and the peritumoral environment. The precise understanding of subpopulations constituting GBM, particularly astrocytes, is not limited to glioma stem cells (GSC) and could help in the understanding of tumor pathophysiology. The anatomical fingerprint is essential for non-invasive assessment of patients’ prognosis and correct surgical/radiotherapy planning.

## 1. Introduction

In the 2021 WHO classification of central nervous system (CNS) tumors, the term glioblastoma (GBM) includes adult-type neoplasms that are among the most common malignant central nervous system (CNS) cancers; they are relatively rare [1,2]. This suggests that the CNS microenvironment is naturally equipped to control proliferative cells, although, rarely, failure of this system can lead to cancer development. Moreover, the adult CNS is innately non-permissive to glioma cell invasion [3].

Recent data suggested that tumor location could predict the biological signature of gliomas, and determine the preferential route of expansion and patients’ prognosis [4].

Despite the fact that many models have been proposed, GBM etiology remains largely unknown [5]. The main excepted hypothesis is that GBM arises from neural stem cells, rather than fully differentiated glia. Neural stem cells are rare, quiescent, self-renewing cells of the central nervous system atop the apex of cellular hierarchies [6]. The adult human CNS has two main niches that harbor neural stem cells (NSC): the subventricular zone (SVZ) and the subgranular zone (SGZ) in the hippocampus [7]. Due to their biological features, especially the capability of self-renewal and migratory potential, NSC might be capable of transformation into glioma stem cells (GSC) [8,9,10]. However, other authors suggest that glial precursors or differentiated astrocytes may also give rise to glioblastomas [11]. Based on traNSCriptional features, three distinct forms of GBM have been revealed (classical, proneural, and meseNSChymal), and the existence of GBM-subtype plasticity has been confirmed [12]. A crucial role in the development and progression of gliomas is held by the isocitrate dehydrogenase (IDH) mutation, which is present in 70% of LGGs [13]. IDH mutation precedes 1p19q co-deletion, which is considered an early event in gliomagenesis. Other genetic modifiers intervene in the definition of the GBM biological signature, resulting in the amplification of the Epidermal Growth Factor Receptor *(EGFR),* loss of heterozygosity or entire loss of chromosome 10, where Phosphatase and tensin homolog (*PTEN*) maps, amplification and mutational activation of receptor tyrosine kinase (RTK) genes, O^6^-methylguanine-DNA methyltransferase (*MGMT*) promoter hypermethylation appear. Mutations in tumor protein TP53 as well as telomerase reverse traNSCriptase (*TERT*) promoter may be also displayed. Copy number alterations in the CDK/RB pathway members axis co-occur [1,14,15]. Oncogenic mutations in NSC affect tumoral migration/infiltration patterns, and a pivotal study [16] has reported the mutational status related to GBM anatomical distribution [16]. Moreover, IDH status predicts distance to the SGZ since the IDH wild-type tumors are more often found near the SGZ. This may imply that IDH wildtype gliomas are of a different anatomical origin than IDH-mutated gliomas [17].

We hypothesize that anatomical factors (CNS regionalized NSC and glial cells, CNS regional extracellular composition, CNS lateralization and committed gene expression) could account for the biological behavior and invasiveness routes of gliomas. Therefore, in the present study, we attempted to find a possible link between the anatomical distribution (tumor core and routes of invasion) of human GBM evidenced by imaging techniques and regional features of the central nervous system, considering markers of the tumor biological signature and well-studied mechanisms/features in the developing and adult human brain.

The precise understanding of GBM etiopathogenesis could be the key to solving this enigma, while the prediction of tumor growth and invasiveness pathways, with system-biology-based models [18], may help to define boundaries of focal treatments such as surgery or radiotherapy.

## 2. Materials and Methods

On 14 February 2022, the PUBMED electronic database was searched, and the following terms were applied: (glioblastoma OR microglia OR extracellular matrix OR neural stem cells OR cancer stem cells AND peritumoral zone). Results were analyzed and processed with the ZOTERO reference manager (Center for History and New Media, George Mason University, Virginia, VA, USA). All papers written in languages other than English were excluded. Time or publication status restrictions were not applied.

## 3. Results

A total of 55 articles were identified using the search algorithm on PUBMED.

### 3.1. Neural Stem Cells

The regional expression of NSC in the SVZ and SGZ could be the differential source of malignancies and was considered as a hypothesis to justify the biological signature of gliomas according to their anatomical distribution [16,17].

NSC are abundant in embryonic neural tissues, and their amount reduces gradually with aging [19]. In the adult human brain, NSC seem to be restricted to the SVZ and the SGZ, preserving their motility to migrate towards specific sites as necessary [19,20]. These relatively quiescent NSC (also known as B1 cells) can be stimulated to become rapidly dividing progenitors (the so-called C cells) that could furtherly differentiate towards A cells (migrating neuronal precursors) [21,22]. Adult neurogenesis plays a pivotal role in maintaining learning capacity and memory. Scientific evidence has highlighted the importance of glial cells and extracellular matrix (ECM) in this context [20]. Within the proliferative embryonic neural tissue, transplanted GSC exhibit reduced proliferation and survival, altered gene expression, and do not form tumors [23]. Likewise, embryonic neural tissue is considered to be non-permissive for GBM cell growth and tumor formation by redirecting differentiation toward a more benign phenotype [24]. Endogenous NSC are recruited to sites of experimental GBM where they exhibit antitumorigenic effects. NSC secrete factors that interfere not only with other stem cells but also with glioma cells. NSC also stimulate the surrounding normal brain tissue to produce various cytokines that can control the proliferation of glioma cells. This recruitment of NSC into GBM tissues occurs efficiently in younger animals, but fewer NSC accumulate in the adult CNS [25,26]. The reason for this hostile microenvironment resides in the proliferative state of NSC that, without robust surveillance mechanisms, could undergo malignant transformation easier than non-proliferative astrocytes or neurons. However, it has been shown that astrocytes and even neurons can undergo oncogenic transformation [27]. Although this evidence suggests a non-permissive microenvironment in NSC niches, aged NSC could have different biological behavior. For instance, NSC release bone morphogenetic protein-7 (BMP7), which downregulates Olig2 in GSC and attenuates their tumorigenicity [28,29]. In the elderly, this intrinsic tumor-suppressor mechanism decreases with the age-related decline in BMP7-secreting NSC [28,29]. The aging-related idea is that senescence might precondition NSC for oncogenic transformation. In fact, at a population level, NSC are less regenerative with age, and rates of neuron differentiation decline in both the SVZ and the hippocampal dentate gyrus [30]. Although a smaller number of cells are observed to be dividing at a given time in the aged forebrain (SVZ), they are more proliferative than young NSC, due to increased rates of cell cycle re-entry [31]. Another important feature of aging NSC (and other tissue-specific stem cells) is their increased tendency towards senescence in response to growth signals [24].

The direction of NSC pathways from the SVZ to the cortex may explain the growth patterns of gliomas (from subcortical areas towards the ventricles). The neurogenesis pathway could be somehow reverted: gliomas may begin in the subcortical sectors, expanding into the subgyral, and then the gyral sectors, towards the lobe and contra-lateral hemisphere following commissural fibers [32]. Alternatively, recent data showed that NSC can also be found in the so-called accessible subcortical white matter and could potentially differentiate into neurons, astrocytes, and oligodendrocytes with a comparable capacity to fetal NSC [33,34]. Either way, the capacity for migration is a pivotal feature to validate the NSC-dependent gliomagenesis hypothesis. Consequently, insight into the biology of progenitor-cell motility in the CNS may lead to a better understanding of the invasion routes of glioma cells, such as the reactivation of migratory genes expressed during CNS development [35].

### 3.2. Embryological Development

The common embryological origin of all CNS subtypes, except the microglia, could reduce the expected contribution of embryogenesis to explain gliomagenesis. The radial glia cells, derived from the neuroepithelium, will produce both neurons and glial cells, maintaining the cellular apicobasal specialization of the cell bodies in the ventricular zone [36]. This is the first regional difference to be noted; although cells appear to divide homogenously, different subtypes of neurons are generated from different areas according to cell-extrinsic positional gradients, such as the dorsoventral (mediolateral) and rostrocaudal axis, through protein expression such as BMP, Wnt and Sonic hedgehog (Shh), and activation of traNSCription factors [37,38]. The differentiation of astrocytes from the radial glia could follow similar positional cues, although the molecular mechanisms have not yet been elucidated. Moreover, the temporal pattern is essential, as from radial glia of the pallium neurogenesis will follow a deep-superficial sequence for projecting neurons, followed by gliogenesis [39]. The temporal cues, contrarily to the spatial gradient, seem to be cell-intrinsic.

Progenitor cells and astrocytes, thus having a potential role in the anatomically related biological signature of gliomas, could retain spatiotemporal diversity.

Inside the SVZ, among NSC, adult B1 cells have been identified as astrocytic cells ensheathing A cell chains (neuronal precursors), together with an ultrastructurally diverse astrocytic population (B2 cells). In particular, the lamina formed by B1 cells separate A cells from the ependyma, while B2 cell chains divided (with some gaps) the A cells from the striatum [22].

B1 cells express glial fibrillary acidic protein (GFAP), glutamate/aspartate transporter (GLAST), and Nestin as astrocytic markers, and their morphology and ultrastructure are comparable to mature differentiated astrocytic cells [22]. B1 fate is not committed towards the neuronal phenotype. Epidermal growth factor (EGF) signaling induces proliferation and migration of B1 cells from the SVZ into the striatum, septum, corpus callosum, and fimbria-fornix, mostly as oligodendrocyte-committed (NG2 progenitors), and oligodendrocytes (pre-myelinating and myelinating) or highly branched GFAP/S100β cells (astrocytic markers) in both striatum and septum, without observable neuronal differentiation [40].

Regional properties seem to be retained by B1 cells. For instance, lateral ganglionic eminences (LGE)-derived lineage expresses traNSCription factors necessary for olfactory bulb interneuron differentiation, probably through a cell-autonomous mechanism. Transplantation of these cells to a heterologous location will not alter their phenotype [41]. Hence, the regional mosaicism of SVZ astrocytes seems to be encoded during development and then maintained.

Regional specificity, together with temporal factors, could potentially influence the growth and invasiveness of mutated clones, partly establishing their migratory routes [42,43].

### 3.3. Astrocytes and the Spatiotemporal Diversity

In the mammalian CNS, astrocytes show inter-regional and intra-regional distinct features during development and adulthood, structuring astrocytic domains, with local regulation of synaptic plasticity [40,44,45]. In mammalian brains, astrocytes are the most abundant cellular component, being classified as protoplasmic and fibrous, according to their morphology, through all the CNS with specialized Bergmann glial cells in the cerebellum and Müller cells in the retina [46]. The formation and maintenance of synaptic circuitry both through gliotransmission and metabolic regulation is indeed a key function of these domains, whose failure could lead to gliopathy [47]. The hominid cortex, compared to the rodent cortex, includes various anatomically defined subclasses of astrocytes.

The interlaminar astrocyte, for instance, copiously resides with their bodies in the superficial cortical layers, under the pial surface, and extends millimetric tortuous (corkscrew-like) long processes without varicosities, terminating in the neuropil and sporadically on the vasculature of cortical layers 3–4 [48]. Another population, typically human, inhabiting layers 5 and 6, extend long processes with characteristic and regularly spaced varicosities. These fibers could allow long-distance signaling across cortical layers or through gray and white matter. Their varicosities also contact blood vessels and could coordinate functional hyperemia across wider areas than a protoplasmic domain [48]. Protoplasmic and fibrous astrocytes are present in both primates and rodents. Human astrocytes are larger and more complex than their murine counterpart but are organized similarly in nonoverlapping (protoplasmic) or overlapping (fibrous) domains [40,47].

The precise anatomical distribution of these morphological subtypes has not yet been investigated relative to the GBM peritumoral microenvironment and could be interesting as they are differently entangled within white matter, cortical layers, and vasculature.

A recent study on a mouse model using a fluorescence-activated cell sorting (FACS)-based strategy identified five astrocyte subpopulations (named with capital letters from A to E) with a particular distribution across the olfactory bulb, brainstem, cerebellum, cortex, and thalamus [49]. Some of these subpopulations were correlated to human glioma (both high- and low-grade). In particular, population B cells were associated with the mesenchymal subtype, worse prognosis, and decreased survival. Population C, on the other hand, was physiologically enriched for genes related to synapse maintenance and formation and, in glioma cells, its prevalence demonstrated a higher occurrence of epilepsy and lower migration potential [49]. Gemistocytes are considered neoplastic GFAP^+^-p65-Bruton’s tyrosine-kinase^+^ (BTK^+^)-expressing astrocytes with a typical cell shape and are found in several brain tumors including GBM [50]. GBM with gemistocytes is characterized by radiological multifocality, with one major lesion surrounded by a few smaller, adjacent lesions, that might be correlated with decreased overall survival. However, the gemistocytic variant does not have clinical significance compared to the non-gesistocytic GBM, and therefore should not be considered as a separated subtype [51]. A better definition of astrocytic subpopulations with higher anatomical resolution and combined with morphologic features could help the prediction of the clinical features of GBM diversity.

### 3.4. Myelin-Associated Signaling

The main molecular inhibitors against glioma invasion have been described in the myelin-associated glial environment. The expression of Nogo isoforms, the Semaphorins/Plexin interaction, and ephrins (although controversial data can be found about their receptors) consistently inhibit tumor progression and invasion [3,52].

ECM-associated proteins of this class are netrins and the slit family, usually required for axon guidance and orientation [53]. The netrins are secreted, laminin-related proteins that are expressed by neurons and glia in the adult CNS. Netrin-1 seems to be fundamental for thalamocortical projection structure in a murine model [54]. In GBM cell lines, Netrin acts in an autocrine manner to inhibit glioma cell motility and promote focal adhesion formation. Netrin-1 is involved in angiogenesis by controlling the proliferation and migration of vascular endothelial cells [55,56].

The slit family of secreted proteins (slit 1–3) includes ECM-associated glycoproteins and ligands for the Roundabout family (Robo 1–4). The slit/Robo pathway is used for the astrocytic tunnel formation that is required for migration and dispersion of new neurons in the adult brain and also inhibits glioma migration and invasion [57]. These mechanisms are widespread throughout the CNS and their failure could explain preferential migration routes and pathways, preferentially through perivascular space and white matter tracts.

### 3.5. Neurons and Glutamate Transporters

Neurons, on the other hand, induce astrocyte maturation via cell–cell contact and the exchange of growth factors [19]. The neuronal membrane is effective in inhibiting astrocyte proliferation, and astrocyte differentiation is mediated by transforming growth factor (TGF)-β signaling, which is initiated by mature neurons. Thus, the interaction between neurons and tumors of glial origin is tumor-suppressive [58]. Adult post-mitotic neurons are capable of killing murine and human gliomas, and cerebellar granular neurons have a more potent killing function than cerebral cortical neurons [59]. This partly explains why cerebellar GBM are quite rare in humans. Neurons in brain tissues adjacent to tumors express program death-ligand-1 (PD-L1), which induces caspase-dependent apoptosis in GBM cells by activating an unknown receptor. In contrast, PD-L1 is also detected in several types of cancer cells, including GBM, and contributes to immunosuppression [59]. Neurons also secrete neurostatins and block cellular proliferation by inhibiting EGFR activation that is necessary for mesenchymal transition. Exposure to neurostatins also enhances the antigenicity of glioma cells through the increased expression of Connexin 43 (Cx43) [60].

Moreover, gliomas release in their defense excessive amounts of glutamate, inducing glutamate excitotoxic neuronal cell death in the peritumoral brain parenchyma, which, in turn, stimulates glial tumor cells by activating glutamate receptors in a paracrine and autocrine manner [61]. Therefore, upregulation of the glutamate transporter-1 (GLT-1) could protect neurons and glia in peritumoral areas by removing extracellular glutamate and directly suppressing tumor cell proliferation GLT-1 expression increases during synaptogenesis and is a marker of astrocyte maturation [62]. To express GLT-1, astrocytes need to contact neurons or endothelial cells. The endothelium is also necessary for astrocytic expression of the GLAST glutamate transporter. The precise signaling has not been established; however, Notch seems to be required [62].

GLAST and GLT-1 together are a significant proportion of the total protein in the brain, representing about 2.1% of proteins in the molecular layer of the cerebellum, 1.6% in the hippocampal stratum radiatum, and 1% of proteins in forebrain tissue, and are by far the most abundant glutamate transporter subtypes in the CNS [63]. The cerebellum and the retina express the highest percentage of GLAST (through the Bergmann and Müller glia) and, together with PD-L1 neuronal expression, could contribute to the relative glioma sparing of the cerebellum [63]. The expression of postnatal human brain tissue has confirmed that GLT1 and GLAST are expressed in all regions, with a higher level in the cerebellar cortex, mediodorsal thalamic nucleus, striatum, amygdala, hippocampus, primary visual cortex, and the dorsal prefrontal cortex [64]. GLAST is in the 99th percentile of expressed proteins in the cerebellar cortex, and 97th percentile in the primary visual and dorsal prefrontal cortices [64]. The different expressions of these transporters in brain regions partly overlap with gliomas prevalence [16].

Different brain regions have a variable composition of neuronal layers (for instance, according to the prevalent afferent/efferent projections) with recognized neuronal layers, although a precise and unanimously accepted taxonomical nomenclature has not been established [65]. The excitatory/inhibitory ratio cannot be the sole element to consider in modulating glioma growth. However, a clear pattern of neuronal arrangement has not been identified to determine how the cell-type prevalence varies among different areas, as was demonstrated, for instance, in the retina [65]. The classification of cerebral cortex cell types has challenged the greatest neuroscientists in history from Ramon y Cajal to Camillo Golgi, who describe neurons with distantly projecting and locally projecting axons. Brodmann and other pioneers of cortical cytoarchitectonic (such as Campbell) gave us the basis for the understanding of functional properties [66]. To add more detail to this puzzle, new neuronal subtypes are emerging coupling traNSCriptomic information to the classical morphological and electrophysiological data [65,67].

### 3.6. Extracellular Matrix and Peritumoral Environment

Hans Joachim Scherer systematically observed, in 1938, that glioma cells migrate through existing brain structures, and pointed out that “…the resemblance of the collections of the glioma cells to the normal architecture is striking”, suggesting a cross-talk between tumoral cells and the parenchyma, involving a “characteristic architectural pattern” [68]. Today, the evidence has reinforced the observational hypothesis that glioma cells could migrate through the perivascular space and white matter tracts [3,69,70]. Nonetheless, growth factors, metabolic coupling and environmental adaptation are essential for GBM growth and invasiveness [71]. Indeed, the ECM is the ultimate gatekeeper that could guide tumor spreading and influence glioma biology in different brain areas throughout the dense network of astrocytes and neurons [72,73].The ECM of the adult CNS is formed by hyaluronic acid and associated glycoproteins. Chondroitin sulfate proteoglycans (CSPG) of the lectican family (aggrecan, versican, neurocan, and brevican) are the major group of hyaluronic acid-binding glycoproteins [72].

This molecular barrier could also constrain tumor cells from invading the brain parenchyma. The dense band of tightly interwoven astrocytic processes and the glycosylated CSPGs that fill the intercellular spaces create a molecular barrier against glioma cell invasion [74]. In contrast, they occasionally promote glioma cell adhesion and migration [75]. Glioma cells that have acquired mesenchymal characteristics can interact with CSPGs and migrate into the normal brain parenchyma through matrix metalloproteinases (MMP) [76,77,78]. Tumor patterns of migration could follow ECM organization. In particular, a non-radial invasion pattern was shown in murine models using specific cellular markers [69]. Tumor cells accumulated in structures located far from the transplant site, such as the optic white matter and pons, whereas certain adjacent regions were spared. As such, the hippocampus was remarkably free of infiltrating tumor cells despite the extensive invasion of surrounding regions [69]. In the same study, clinical verification was performed, and most malignant temporal lobe gliomas were located lateral to the collateral sulcus. Despite widespread pathological fluid-attenuated inversion recovery (FLAIR) signal in the temporal lobe, 74% of “lateral tumors” did not show signs of involvement of the amygdala-hippocampal complex [69] (Figure 1).

Another study, mapping the normal expression of ECM proteins, showed a particular asset of aggrecan, which interestingly matched the blocked routes of migration (hippocampal inlet, amygdala, and hypothalamus), while tenascin-R was largely expressed along the routes of preferred migration such as commissural fibers, the lateral portion of the hippocampus [79]. This speculation needs to be supported by data on tumoral models, since these results were obtained under normal conditions [79]. However, the unique and complex ECM distribution, fundamental for brain physiology, represents an intriguing field of investigation to assess particular markers of gliomas. Encoded distribution of distinct ECM proteins could explain routes of migration and the biological signature of gliomas.

Microglia and macrophages are both the most expressed cytotypes in the tumoral and peritumoral microenvironment (almost one-third of the tumor mass) and among the major modifiers of the ECM [71,80,81]. The link between microglia and tumoral cells, involving ECM remodeling, was first suggested by Sir Wilder Penfield [82]. The microglia-specific polarization phenotype (M1, M2) or transitional dynamic states of permissive/non-permissive behavior could hold the balance of tumor growth and invasiveness, but, to the best of our knowledge, no data show regionality of these phenomena.

### 3.7. Atlas of GBM and the Lateralization Puzzle

Accurate anatomical profiling of GBM can be made using volumetric magnetic resonance imaging (MRI) studies. From a topographical point of view, we can identify different types of GBM. The boundaries of GBM are usually ill defined but the anatomical origin of the bulk could limit their invasiveness along specific routes, compartmentally, rarely trespassing neighboring areas [32,69]. In particular, these tumors could be subdivided into: gyral GBM, subgyral GBM (Figure 2), and a lobar GBM with the possibility, in this case, to involve commissural fibers (Figure 3).

Esmaeili et al. on the other hand, recently showed a radiological series of patients whose GBM invasiveness was not anatomically confined and tended to follow white matter routes (Figure 4) traced with a diffusion tensor imaging (DTI) atlas [70]. Therefore, as part of the tumor’s biological signature, GBM could express an anatomically restricted bulky morphology or diffusive phenotype.

In clinical practice, the ill-defined margin of GBM invasiveness is the major challenge. Prediction of tumor growth pathways could allow redefining focal treatments such as surgery or radiotherapy. The evidence suggests, simplistically, at least two theoretical behaviors of the GSC relative to the environment: (a) periventricular bulky growth partly limited inside a triangular section having the base in the gyrus and apex pointing to the ventricle wall; (b) a diffusive pattern that follows the white matter fascicles [83,84].

Jang et al., in a retrospective analysis of 585 surgical cases, tried to measure this behavior using MRI parameters: a ratio between volumes obtained through FLAIR sequences and contrast-enhanced T1-weighted images (CE-T1WI). FLAIR and CE-T1WI volumes were used as indirect indices of diffusivity and tumor bulk respectively. Hence, a cut-off value of the ratio was established to categorize the GBM as proliferative- or diffusion-dominant phenotypes [85] (Figure 5).

A significant correlation was found between anatomical distribution of gliomas, biological behavior, and patient prognosis, reinforcing the idea of a region-specific biological signature (considering both tumorigenesis and invasiveness) [4,16,17,86,87].

An attempt to build an atlas of GBM phenotypes showed that most tumors grow into the periventricular white matter regions adjacent to the SVZ [16] (Figure 4). Moreover, it was shown that the age of the patient, left-right lateralization, and temporal or frontal localization of the tumor were significantly correlated to distinct biological signatures. Furthermore, the proneural versus mesenchymal phenotypes, and consequently aggressiveness of the tumors, showed a regional distribution [16]. While many features observed in the probabilistic radiographic atlas of GBM could be speculatively explained by the aforementioned mechanisms, the lateralization remains puzzling indeed. Recently, Fyllingen et al. demonstrated on a retrospective analysis of 215 patients that there are differences in overall survival (OS) of patients based on GBM location that are not limited to eloquence. OS was shorter in patients with centrally located tumors (corpus callosum and basal ganglia) and tumors in the left temporal lobe pole, and higher in patients with tumor location in the right dorsomedial temporal lobe and white matter region involving the left anterior paracentral gyrus/dorsal supplementary motor area/medial precentral gyrus [88].

A pivotal study demonstrated human cortical asymmetry considering gene expression levels between the left and right embryonic hemispheres [89]. The authors identified and verified 27 differentially expressed genes, which suggests that human cortical asymmetry is accompanied by early, marked traNSCriptional asymmetries. As regards the LIM domain, only protein 4 (LMO4) in particular is consistently more highly expressed in the right perisylvian human cerebral cortex than in the left. Although essential for cortical development, also in mice, Lmo4 expression was moderately asymmetrical in every individual brain and was not consistently lateralized to the right or left side. This may relate to behavioral and anatomical studies in mice, in which sensory-motor asymmetries, such as paw preference, are observed in individual mice but not on a population level. Right or left hemisphere dominance, as well as hand preference, is peculiarly human and reflects early stage asymmetric cortical development [89].

Moreover, LMO4 was identified as an essential cofactor in Snail2-mediated cadherin repression and the epithelial-to-mesenchymal transition of both neural crest and neuro-blastoma cells. Eventually, the reactivation of lateralized progenitors expressing LMO4 could regulate cadherin expression and, hence, tumor invasiveness [90].

### 3.8. Surgical Implications

The knowledge of regional anatomy associated with the precise understanding of glioma biology remains a mainstay to plan the correct operation. Many surgeons, bypassing operation planning, execute an intralesional piecemeal resection in which the tumor is removed from the center toward the edges. This technique does not require any anatomical orientation and it is considered safe because surgery remains within the tumor-avoiding healthy tissue. Nevertheless, a cornerstone of general surgical oncology is to avoid violating tumor margins. In this way, it is crucial to study the anatomical correlation between tumor and healthy brain tissue with special attention to the eloquent cortical and subcortical areas [83,84]. The perfect understanding of tumor spread and diffusion allows the identification of the precise segment that should be disconnected and resected. Sawaya et al. suggest that an en bloc circumferential resection anatomically guided by sulci (in which the edges of the tumor are defined at the beginning of the resection by using the surrounding sulci as a guide) is independently associated with statistically significantly higher rates of complete resection, without an increase in neurological complications [88,91]. A clinical limitation of the present paper is that the recent WHO classification [1] cannot be taken into account for the analysis of the literature. The clinical evidence and animal models refer to prior classification methods. However, despite a lot of discussed radiological and biological differences, GBM studies and clinical data showed similar key aspects that we considered in our speculative assessment of the published evidence.

## 4. Conclusions

The biological factors such as NSC and GSCs, neuronal and glial subtypes, lateralized gene expression, spatiotemporal-committed developmental programs, and extracellular composition could explain the anatomical distribution of GBM and predict the tumor invasiveness routes.

The anatomical fingerprint is essential for the non-invasive assessment of patients’ prognosis and response to specific surgical or radiotherapeutic treatments. Therefore, we attempt to speculate a possible link between the anatomical spreading of imaged-human-GBM and regional features of the central nervous system, considering markers of the tumor biological signature and known mechanisms/features in the developing and adult human brain. The main findings are summarized in Table 1. Based on these assumptions, a classification that integrates the morphological, topographical, and biomolecular data sets with a computer-assisted system-biology approach, could be useful to obtain information for gliomagenesis studies, prognostic and therapeutic purposes. The future direction is the *morphomics* of GBM: morphological and anatomical analysis combined to omics data with a single-cell deterministic approach. The biological signature should be integrated with anatomical and clinical evidence to obtain more reliable understanding of GBM pathophysiology.

## Figures and Tables

**Figure 1 cells-11-01349-f001:**
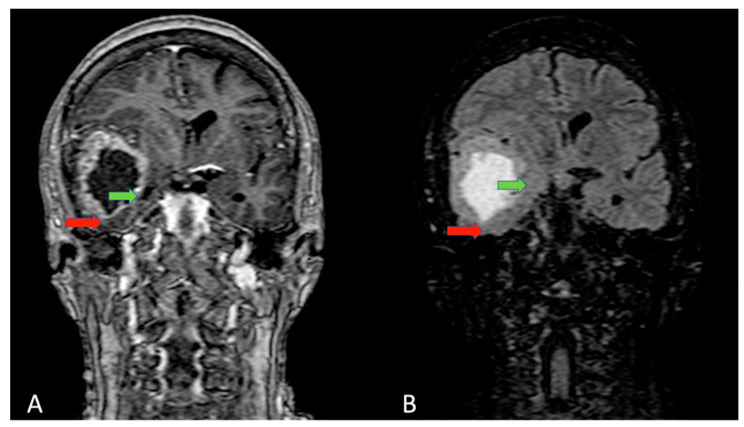
Coronal section of an MRI T1 with gadolinium (**A**) and FLAIR sequences (**B**). GBM arises and grows into the lateral compartment of the right temporal lobe (middle temporal gyrus), compressing and dislocating the superior temporal gyrus and sparing the inferior temporal gyrus, fusiform gyrus (red arrow), and parahippocampal gyrus (green arrow).

**Figure 2 cells-11-01349-f002:**
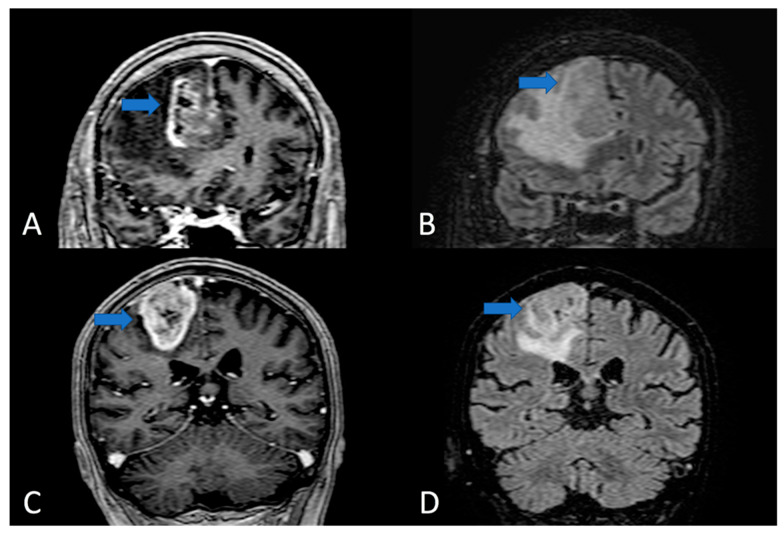
Coronal section of MRI T1 with gadolinium (**A**,**C**) and FLAIR sequences (**B**,**D**). In A and B images, the GBM mass located in the subgyral sector developed into superior frontal gyrus/cyngulum. The enhancing nodule is confined in a precise anatomical sector and apparently does not invade the superior frontal sulcus (blue arrow). In (**C**,**D**), a gyral GBM that is completely confined in the ascending parietal gyrus (the blue arrow shows the spared sulcus delimiting the tumor).

**Figure 3 cells-11-01349-f003:**
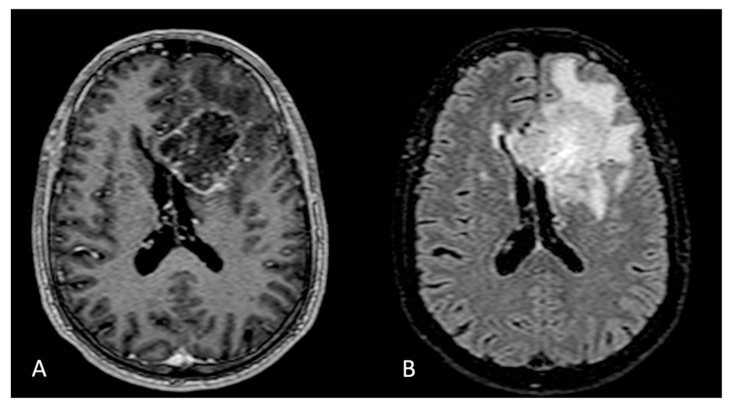
Axial section of MRI T1 with gadolinium (**A**) and FLAIR sequences (**B**). Left frontal lobe GBM with involvement of the corpus callosum.

**Figure 4 cells-11-01349-f004:**
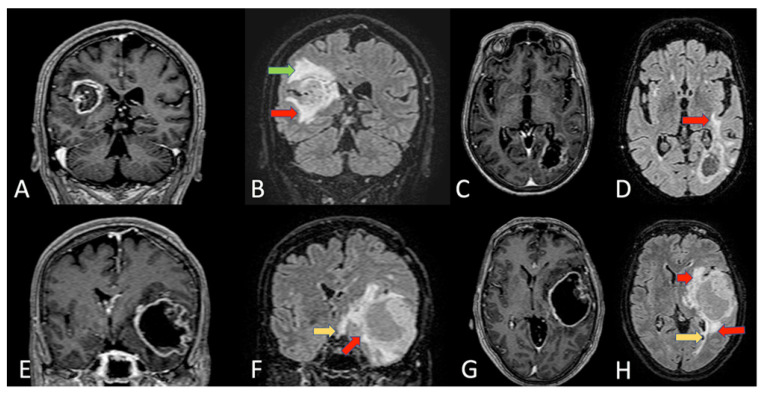
Coronal section of MRI T1 with gadolinium (**A**) and FLAIR sequences (**B**) of a right paratrigonal GBM. In this case, there are clear anatomical relationships between the tumor and lateral wall of the atrium. In the FLAIR sequence, the infiltration of the superior longitudinal fascicle (green arrow) and the inferior fronto-occipital fascicle (IFOF) (red arrow) are visible. In (**C**,**D**), we can see an axial section of a T1 with gadolinium (**C**) and FLAIR sequences (**D**) of a left paratrigonal GBM. Red arrow (**D**) indicates the IFOF tumoral infiltration at the level of the external capsule. In the lower panels, there is a coronal section of MRI T1 with gadolinium (**E**) and FLAIR sequences (**F**) of a left temporal GBM. In F, the tumoral infiltration of all temporal lobe, of the external capsule, of the IFOF (red arrow) and the omolateral optic tract (yellow arrow) are visible. Axial section of the same patient MRI T1 with gadolinium (**G**) and FLAIR sequences (**H**). Red arrows (**H**) indicate the IFOF tumoral infiltration and yellow arrows indicate the infiltration of optic radiation.

**Figure 5 cells-11-01349-f005:**
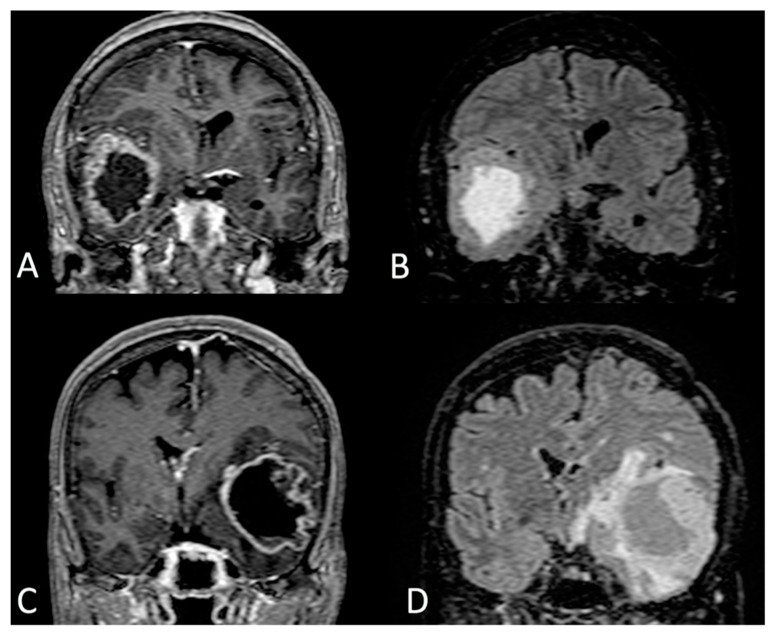
Coronal section of MRI T1 with gadolinium (**A**) and FLAIR sequences (**B**) of a proliferation-dominant type of temporal GBM (bulky tumor with compressive effect and a low rate of peritumoral infiltrative areas). Coronal section of MRI T1 with gadolinium (**C**) and FLAIR sequences (**D**) diffusion-dominant type of a temporal GBM (bulky tumor with extensive infiltration of white matter in peritumoral areas).

**Table 1 cells-11-01349-t001:** Cellular and extracellular elements with the putative molecular mechanisms that could determine the biological signature of GBM, according to a specific anatomical localization. ECM: extracellular matrix; GSC: glioma stem cell; NSC: neural stem cell; SGZ: subgranular zone; SVZ: subventricular zone.

Cellular and Extracellular Elements	Anatomical Localization	Putative Mechanism	Ref.	GBM Biological Signature
NSC	SVG, SGZ	Age-related decline of the tumor-suppressor BMP7	[24,28,29,30]	GBM growth
NSC or progenitor cells	Subcortical white matter	Reactivation of migratory genes (EGF, Lck)	[35,92]	GBM invasiveness
Human ectodermic progenitors	Cortex	Differentially expressed genes (i.e., NEUROD6, ID2, LMO4)	[89,90]	GBM lateralization
Developing Astrocytes (B1 cells)	SVZ, Cortex	EGF overexpression	[42,43]	GBM invasiveness
Astrocytic subpopulations	Thalamus, Cortex, Brainstem	Mesenchymal signature (cluster B); Epilepsy-associated genes enrichment (cluster C)	[49]	GBM invasiveness
Astrocytes	Cerebellum, primary visual and dorsal prefrontal cortices	GLAST/GLT-1 overexpression	[16,63,64,93]	GBM growth (spared regions)
Neurons	Cerebellum	PD-L1 overexpression. Neurostatin release.	[59,60]	GBM growth (spared regions)
Oligodendrocytes	White matter	Nogo, semaphorin, ephrins downregulation	[3,52]	GBM invasiveness
Extracellular matrix	Hippocampal inlet, amygdala, and hypothalamus	Particular asset of aggrecan expression in contrast to Tenascin-R.	[69,79]	GBM growth (spared regions)
GSC	Left temporal lobe	MGMT methylated promoter; EGFR amplification	[16,86]	Bulky phenotype (short overall survivor)
GSC	Right temporal lobe	MGMT unmethylated promoter; IDH1 WT; Mesenchymal signature	[16]	Diffusive phenotype (short overall survival)
GSC	Frontal lobe	Focal PTEN loss; IDH1^R132mut^;p53^mut^; Focal EGFR amplification; Proneural signature	[16,87]	Diffusive phenotype

## Data Availability

Not applicable.

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
