# Peer review of "Regional Development of Glioblastoma: The Anatomical Conundrum of Cancer Biology and Its Surgical Implication"

_cells, 2022, doi:10.3390/cells11081349_

Round 1
Reviewer 1 Report
This study attempted to use existing published work to examine the hypothesis that regional differences in neural stem cells (NSCs) could be the reason glioma cancers are heterogfenious with regards to growth and invasion. I am not sure that the work answered the hypothesis. Instead it reviewed a number of aspects of glioma genesis and the relationship of gliomas with NSCs. It should have started with a concise paragraph stating explicitly that gliomas are caused by genetic mutations in one of there gene axes, p53, RB and RTKs. These need not occur in the cell of origin but anywhere along the lineage from which gliomas arise. The manuscript contains useful information for the student of gliomas and possibly some "food for thought". As such it is a valuable addition. There are some statements that ar not supported by the literature as far as I know and no citations are provided. This is particularly true for page 5, 3.4....Adult post-mitotic neurons are capable of killing murine and human gliomas, ...not aware of this. The opposite is certainly the case.Author Response
REV. 1
Comments and Suggestions for Authors
This study attempted to use existing published work to examine the hypothesis that regional differences in neural stem cells (NSCs) could be the reason glioma cancers are heterogeneous with regard to growth and invasion. I am not sure that the work answered the hypothesis. Instead, it reviewed a number of aspects of gliomagenesis and the relationship of gliomas with NSCs. It should have started with a concise paragraph stating explicitly that gliomas are caused by genetic mutations in one of their gene axes, p53, RB, and RTKs. These need not occur in the cell of origin but anywhere along the lineage from which gliomas arise. The manuscript contains useful information for the student of gliomas and possibly some "food for thought". As such it is a valuable addition.
With the present study, we attempt to speculate a possible link between the anatomical spreading of GBM and regional features of the central nervous system, considering markers of the tumor biological signature and known mechanisms in the developing and adult human brain. We improved the structure of the review to make it clearer.
As suggested by the reviewer, we also introduced some lines regarding the major gene signature for glioblastomas (please see the Introduction).
There are some statements that are not supported by the literature as far as I know and no citations are provided. This is particularly true on page 5, 3.4...Adult post-mitotic neurons are capable of killing murine and human gliomas, ...not aware of this. The opposite is certainly the case.
We read carefully the manuscript and we added new references throughout the text, in particular for “Adult post-mitotic neurons are capable of killing murine and human gliomas” the ref is “Liu, Y.; Carlsson, R.; Ambjørn, M.; Hasan, M.; Badn, W.; Darabi, A.; Siesjö, P.; Issazadeh-Navikas, S. PD-L1 Expression by Neurons Nearby Tumors Indicates Better Prognosis in Glioblastoma Patients. J. Neurosci. 2013, 33, 14231–14245, doi:10.1523/JNEUROSCI.5812-12.2013.”.
Reviewer 2 Report
The authors present a review of the current literature on the biological properties of high-grade gliomas in correlation to the anatomical localization in the brain. The main conclusions are that 1) the biological diversity of gliomas is explained by a combination of anatomical factors such as neuronal and glial subtypes surrounding the tumor, lateralized gene-expression, spatiotemporal committed stem cells, and extracellular composition, and 2) further understanding of these factors could predict the invasiveness of these tumors.
The paper is well written and interesting from a biological as well as clinical point of view. Invasion of gliomas constitutes one of the major obstacles for effective therapy, and more knowledge on this topic is undoubtedly needed. However, there are a few concerns:
- The term high-grade glioma is not accurate. The authors should define the tumor type more accurately by using the WHO 2021 classification. Ideally, the review should be restricted to “classical glioblastomas”, i.e. IDH-wildtype GBM. Since these tumors constitute a clear separate entity, the reported findings will be much more relevant. I appreciate that there may be some methodological problems, since most of the literature will include mixtures of histopathological diagnoses, but at least the tumor subtypes should be specified when referring to the different studies.
- The hypothesis that “anatomical factors (regionalized glial cells, extracellular composition, committed gene-expression) could account for the biological behavior and invasiveness routes of gliomas” should be strictly followed throughout the paper. The NSCs are of relevance in this respect, because of the specific anatomical localization of these cells, but the discussion on the origin of gliomas and of cancer stem cells (CSC) (under “neural stem cells”) is out of place and confusing. If the authors want to discuss the role of CSC in gliomas in this review, they should provide a definition of what is meant by CSC (please see Lathia et al, Genes & Development 2015) and explain more clearly how this complex topic is related to the hypothesis of their work.
- A table (or diagram) that summarizes the main findings (i.e. the relevance of the different parameters in correlation to the anatomical localization in the brain) would be welcome. I find the lateralization concept very interesting, and would suggest that these findings are highlighted in a table/figure.
- As for the conclusions, I would appreciate one or two (more speculative) sentences with suggestions for how to move forward in this area and the focus of future studies.
Author Response
REV 2
Comments and Suggestions for Authors
The authors present a review of the current literature on the biological properties of high-grade gliomas in correlation to the anatomical localization in the brain. The main conclusions are that 1) the biological diversity of gliomas is explained by a combination of anatomical factors such as neuronal and glial subtypes surrounding the tumor, lateralized gene expression, spatiotemporal committed stem cells, and extracellular composition, and 2) further understanding of these factors could predict the invasiveness of these tumors.
The paper is well written and interesting from a biological as well as clinical point of view. Invasion of gliomas constitutes one of the major obstacles to effective therapy, and more knowledge on this topic is undoubtedly needed. However, there are a few concerns:
1. The term high-grade glioma is not accurate. The authors should define the tumor type more accurately by using the WHO 2021 classification. Ideally, the review should be restricted to “classical glioblastomas”, i.e. IDH-wildtype GBM. Since these tumors constitute a clear separate entity, the reported findings will be much more relevant. I appreciate that there may be some methodological problems since most of the literature will include mixtures of histopathological diagnoses, but at least the tumor subtypes should be specified when referring to the different studies.
As suggested by the reviewer, we restricted the search criteria for GBM in our study. However, the papers we considered do not include only the classic GBM (IDH WT), due to the methodological limits that we discussed.
2. The hypothesis that “anatomical factors (regionalized glial cells, extracellular composition, committed gene-expression) could account for the biological behavior and invasiveness routes of gliomas” should be strictly followed throughout the paper. The NSCs are of relevance in this respect, because of the specific anatomical localization of these cells, but the discussion on the origin of gliomas and of cancer stem cells (CSC) (under “neural stem cells”) is out of place and confusing. If the authors want to discuss the role of CSC in gliomas in this review, they should provide a definition of what is meant by CSC (please see Lathia et al, Genes & Development 2015) and explain more clearly how this complex topic is related to the hypothesis of their work.
With the present study, we attempt to speculate a possible link between the anatomical spreading of GBM and regional features of the central nervous system, considering markers of the tumor biological signature and known mechanisms/features in the developing and adult human brain which include the potential of the cancer stem cells, that we defined glioma stem cells (GSC). We re-organized the structure of the review to make it clear and we provided the definitions of NSCs and GSCs.
3. A table (or diagram) that summarizes the main findings (i.e. the relevance of the different parameters in correlation to the anatomical localization in the brain) would be welcome. I find the lateralization concept very interesting and would suggest that these findings are highlighted in a table/figure.
We thank the reviewer for the relevant suggestion. We summarized the main findings in a table. We highlighted the possible link between the parameters discussed in the text and the hypothesis on GBM-related features (please see Table1).
4. As for the conclusions, I would appreciate one or two (more speculative) sentences with suggestions for how to move forward in this area and the focus of future studies.
We added two speculative sentences in the conclusion to indicate the direction of pioneering future studies.
Round 2
Reviewer 1 Report
The manuscript gained further clarity in its revision.
Reviewer 2 Report
The authors have adequately addressed the different comments. Apart from some typos (f.ex. non-gesmistocytic; traNSCription. Also, please check the first sentence of the Introduction) I find the paper suitable for publication.